# The modulation of neural gain facilitates a transition between functional segregation and integration in the brain

James M Shine[1,2]*, Matthew J Aburn[3], Michael Breakspear[3,4], Russell A Poldrack[1]*

[1]Department of Psychology, Stanford University, Stanford, United States; [2]Central Clinical School, The University of Sydney, Sydney, Australia; [3]QIMR Berghofer Medical Research Institute, Brisbane, Australia; [4]Metro North Mental Health Service, Brisbane, Australia

**Abstract** Cognitive function relies on a dynamic, context-sensitive balance between functional integration and segregation in the brain. Previous work has proposed that this balance is mediated by global fluctuations in neural gain by projections from ascending neuromodulatory nuclei. To test this hypothesis in silico, we studied the effects of neural gain on network dynamics in a model of large-scale neuronal dynamics. We found that increases in neural gain directed the network through an abrupt dynamical transition, leading to an integrated network topology that was maximal in frontoparietal 'rich club' regions. This gain-mediated transition was also associated with increased topological complexity, as well as increased variability in time-resolved topological structure, further highlighting the potential computational benefits of the gain-mediated network transition. These results support the hypothesis that neural gain modulation has the computational capacity to mediate the balance between integration and segregation in the brain.
DOI: https://doi.org/10.7554/eLife.31130.001

**\*For correspondence:**
mac.shine@sydney.edu.au (JMS);
poldrack@stanford.edu (RAP)

**Competing interests:** The authors declare that no competing interests exist.

## Introduction

The function of complex networks such as the human brain requires a trade-off between functional specialization and global communication (*Deco et al., 2015a*; *Park and Friston, 2013*; *Tononi et al., 1994*). Contemporary models of brain function suggest that this balance is manifest through dynamically changing patterns of correlated activity, constrained by the brains' structural backbone (*Deco et al., 2013*; *Honey et al., 2007*; *Varela et al., 2001*). This in turn allows exploration of a repertoire of cortical states that balance the opposing topological properties of segregation (i.e. modular architectures with high functional specialization) and integration (i.e. interconnection between specialist regions [*Deco et al., 2015b*; *Ghosh et al., 2008*]).

Recent work has demonstrated that the extent of integration in the brain is important for a range of cognitive functions, including effective task performance (*Bassett et al., 2015*; *Shine et al., 2016a*), episodic memory retrieval (*Westphal et al., 2017*) and conscious awareness (*Barttfeld et al., 2015*; *Godwin et al., 2015*). Furthermore, the topological properties of functional brain networks have been shown to fluctuate over time (*Chang and Glover, 2010*; *Hutchison et al., 2013*), both within individual neuroimaging sessions (*Shine et al., 2016a*; *Zalesky et al., 2014*) and over the course of weeks to months (*Shine et al., 2016b*). While the extent of integration in the brain may relate to more effective inter-regional communication, perhaps via synchronous oscillatory activity (*Fries, 2015*; *Lisman and Jensen, 2013*; *Varela et al., 2001*), there are also benefits related to a relatively segregated network architecture, including lower metabolic costs (*Bullmore and Sporns, 2012*; *Zalesky et al., 2014*) and effective performance as a function of learning

(*Bassett et al., 2015*). However, despite these insights, the biological mechanisms responsible for driving fluctuations between integration and segregation remain unclear.

A candidate mechanism underlying flexible brain network dynamics is the global alteration in neural gain mediated by ascending neuromodulatory nuclei such as the locus coeruleus (*Aston-Jones and Cohen, 2005*; *Sara, 2009*). This small pontine nucleus projects diffusely throughout the brain and releases noradrenaline, a potent modulatory neurotransmitter that alters the precision and responsivity of targeted neurons (*Waterhouse et al., 1988*). Alterations in this system are known to play a crucial role in cognition, as there is evidence for a nonlinear (inverted-U shaped) relationship between noradrenaline concentration and cognitive performance (*Robbins and Arnsten, 2009*; *Figure 1a*).

Mechanistically, the noradrenergic system has been shown to alter neural gain (*Servan-Schreiber et al., 1990*) *Figure 1b*), increasing the signal to noise ratio of afferent input onto regions targeted by projections from the locus coeruleus. A crucial question is how these local changes in neural gain influence the configuration of the brain at the network level. Recent work has linked fluctuations in network topology to changes in pupil diameter (*Eldar et al., 2013*; *Shine et al., 2016a*; *Shine et al., 2018*), an indirect measure of locus coeruleus activity (*Joshi et al., 2016*; *Murphy et al., 2014*; *Reimer et al., 2014*, *2016*), providing evidence for a link between the noradrenergic system and network-level topology. However, despite these insights, the mechanisms through which alterations in neural gain mediate fluctuations in global network topology are poorly understood.

Biophysical models of large-scale neuronal activity have yielded numerous insights into the dynamics of brain function, both during the resting state as well as in the context of task-driven brain function (*Deco et al., 2009*; *Honey et al., 2007*); for review, see *Breakspear, 2017*. Whereas prior research in this area has examined the influence of local dynamics, coupling strength, structural network topology and stochastic fluctuations on functional network topology (*Deco et al., 2015b*; *Deco and Jirsa, 2012*; *Deco et al., 2017*; *Gollo et al., 2015*; *Woolrich and Stephan, 2013*), the direct influence of neural gain has not been studied. Here, we used a combination of biophysical modeling and graph theoretical analyses (*Sporns, 2013*) to characterize the effect of neural gain on emergent network topology. Based on previous work (*Shine et al., 2016a*; *Shine et al., 2018*), we

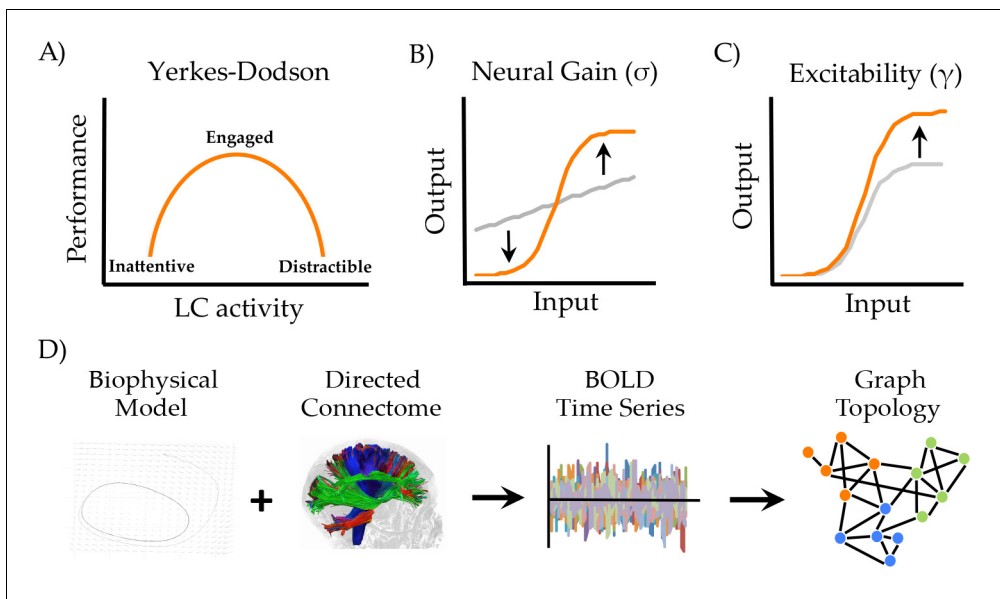

**Figure 1.** Manipulating neural gain. (a) the Yerkes-Dodson relationship linking activity in the locus coeruleus nucleus to cognitive performance; (b) neural gain is modeled by a parameter (σ) that increases the maximum slope of the transfer function between incoming and outgoing activity within a brain region; (c) excitability is modeled by a parameter (γ) that amplifies the level of output; (d) the approach presently used to estimate network topology from the biophysical model.
DOI: https://doi.org/10.7554/eLife.31130.002

hypothesized that manipulations of neural gain would modulate the extent of integration in time-averaged patterns of functional connectivity.

## Results

To test this hypothesis, we implemented a generic 2-dimensional neuronal oscillator model (*Fitzhugh, 1961*; *Stefanescu and Jirsa, 2011*) within the Virtual Brain toolbox (*Jirsa et al., 2010*; *Sanz Leon et al., 2013*) to generate regional time series that were constrained by a directed white matter connectome derived from the CoCoMac database (*Kötter, 2004*) *Figure 1d*). The simulated neuronal time series were passed through a Balloon-Windkessel model to simulate realistic BOLD data. Graph theoretical analyses were then applied to time-averaged correlations of regional BOLD data to estimate the functional topological signatures of network fluctuations (see Materials and methods for further details).

To simulate the effect of ascending neuromodulatory effects on inter-regional dynamics, we systematically manipulated neural gain (σ; *Figure 1b*) and excitability (γ; *Figure 1c*). These two parameters alter different aspects of a sigmoidal transfer function, which models the nonlinear relationship between presynaptic afferent inputs and local firing rates (*Freeman, 1979*). When the σ and γ parameters are both low, fluctuations in regional activity arise mainly due to noise and local feedback. As the σ and γ parameters increase, the influence of activity communicated from connected input regions also increases, leading to non-linear cross-talk and hence, changes in global brain topology and dynamics. Here, we investigated the topological signature of simulated BOLD time series across a parameter space spanned by σ and γ in order to understand the combined effect of neural gain and excitability on global brain network dynamics.

### Neural gain and excitability modulate network-level topological integration

We simulated BOLD time series data across a range of σ (0–1) and γ (0–1) and then subjected the time series from our simulation to graph theoretical analyses (*Rubinov and Sporns, 2010*). This allowed us to estimate the amount of integration in the time-averaged functional connectivity matrix across the parameter space (*Figure 2a*). Specifically, we used the mean participation coefficient ($B_A$) of the time-averaged connectivity matrix at each combination of σ and γ. High values of mean $B_A$ suggest a relative increase in inter-modular connectivity, thus promoting the diversity of connections between modules (*Bertolero et al., 2017*) and increasing the integrative signature of the network (*Shine et al., 2016a*). The converse situation (i.e., segregation) can thus be indexed by low mean $B_A$ scores, or alternatively by the modularity statistic, $Q$. We observed a complex relationship between σ, γ and $B_A$, such that maximal integration occurred at high levels of σ but with intermediate values of γ. Outside of this zone, the time-averaged connectome was markedly less integrated. Similar patterns were observed for other topological measures of integration, such as the inverse modularity ($Q^{-1}$) and global efficiency (*Figure 2—figure supplement 1*).

### Neural gain transitions the network across a critical boundary

The relative simplicity of our local neural model allows formal quantification of the inter-regional phase relationships that characterize the underlying neuronal dynamics. These fast neuronal phase dynamics compliment the view given by the slow BOLD amplitude fluctuations and give insight into their fundamental dynamic causes. We employed a phase order parameter, that quantifies the extent to which regions within the network align their oscillatory phase – high values on this scale reflect highly ordered synchronous oscillations across the network, whereas low values reflect a relatively asynchronous system (*Breakspear et al., 2010*; *Kuramoto, 1984*).

Across the parameter space, we observed two clear states (*Figure 2b*): one associated with high (ρ ≥0.5; yellow) and one with low (ρ <0.5; blue) mean synchrony, with a clear critical boundary demarcating the two states (dotted white line in *Figure 2a/b*) that was associated with a relative increase in the standard deviation of the order parameter (*Figure 2—figure supplement 2a*). This strong demarcation between states is a known signature of critical behavior (*Chialvo, 2010*), which can occur at both the regional and network level. We observed evidence for both regional and network criticality in our simulation, whereby small changes in parameters (here, σ and γ) facilitated an abrupt transition between qualitatively distinct states. At the regional level, this pattern is observed

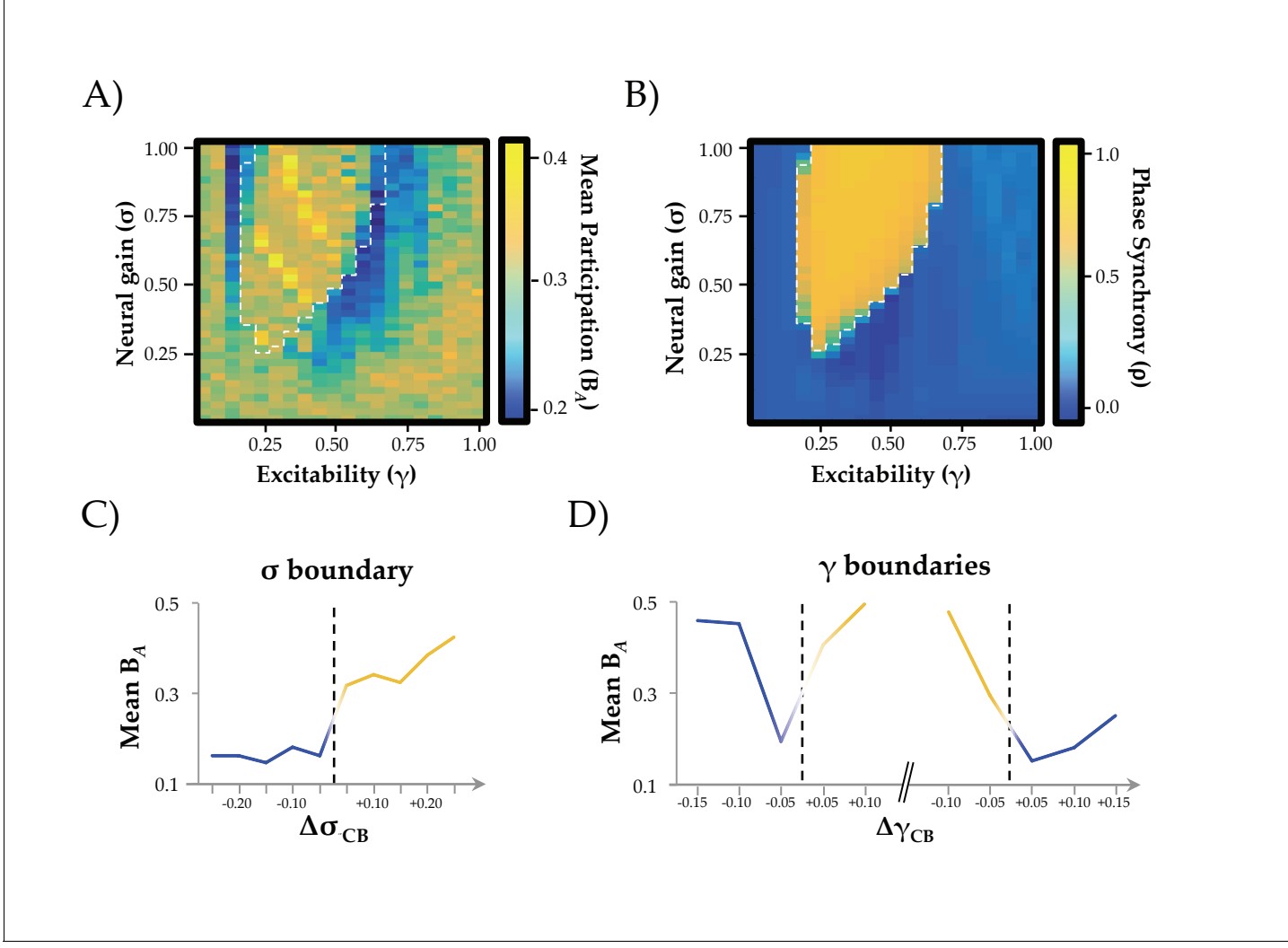

**Figure 2.** Network Integration and Phase Synchrony. (a) mean participation as a function of σ and γ; (b) phase synchrony (ρ) as a function of σ and γ; (c) mean participation ($B_A$) aligned to the critical point (represented here as a dotted line) as a function of increasing σ; (d) $B_A$ aligned to the critical point as a function of increasing γ – the left and right dotted lines depicts the synchrony change at low and high γ, respectively. The *y*-axis in (c) and (d) represents the distance in parameter space aligned to the critical point/bifurcation for either σ ($\Delta\sigma_{CB}$; mean across $0.2 \leq \gamma \leq 0.6$) or γ ($\Delta\gamma_{CB}$; mean across $0.3 \leq \sigma \leq 1.0$). Lines are colored according to the state of phase synchrony on either side of the bifurcation (blue: low synchrony; yellow: high synchrony).

DOI: https://doi.org/10.7554/eLife.31130.003

The following figure supplements are available for figure 2:

**Figure supplement 1.** Relationship between phase regimen boundary and alternative measures of network integration.

DOI: https://doi.org/10.7554/eLife.31130.004

**Figure supplement 2.** Standard deviation of the order parameter across the parameter space.

DOI: https://doi.org/10.7554/eLife.31130.005

**Figure supplement 3.** Transition to self-sustained oscillations in a single brain region.

DOI: https://doi.org/10.7554/eLife.31130.006

**Figure supplement 4.** Average time-averaged connectivity matrix in regions of the parameter space associated with high (yellow) or low (blue) ordered phase synchrony.

DOI: https://doi.org/10.7554/eLife.31130.007

as a transition from input-driven fluctuations about a stable equilibrium to self-sustained oscillations (*Figure 2—figure supplement 3*). At the network level, the combined influence of increased gain

and structural connections manifest as a transition to high amplitude, inter-regional phase synchrony (*Figure 2—figure supplement 2b*).

To further disambiguate the system-level dynamics, we studied the probability distribution of the fluctuations in the order parameter. Close to the boundary, we observed a truncated Pareto (i.e., power law) scaling regime, spanning up to two orders of magnitude (*Figure 2—figure supplement 2b*). This pattern is consistent with a critical bifurcation within a complex system consisting of many components (see *Cocchi et al., 2017* and *Heitmann and Breakspear, 2017*Heitman and Breakspear, 2017 for further discussion). After crossing the boundary, this relationship develops a 'knee' above the power-law scaling (*Figure 2—figure supplement 2b*), consistent with the emergence of a characteristic temporal scale in a super-critical system (*Roberts et al., 2015*). These observations suggest that the system undergoes a bifurcation across a critical boundary as the synchronization manifold loses stability.

A host of contemporary neuroscientific theories hypothesize that temporal phase synchrony between regions underlies effective communication between neural regions (*Fries, 2015*; *Lisman and Jensen, 2013*; *Varela et al., 2001*), which would otherwise remain isolated if not brought into temporal lockstep with one another. As such, we might expect that the changes in neural gain that integrate the brain might do so through the modulation of inter-regional phase synchrony. Our results were consistent with this hypothesis. By aligning changes in the topological signature of the network to the critical point delineating the two states, we were able to demonstrate a significant increase in integration (mean $B_A$; $T_{798} = 2.57$; p=0.01) and decrease in segregation (Q; $T_{798} = -17.44$; p<0.001) of network-level BOLD fluctuations in the highly phase synchronous state. Specifically, global integration demonstrated a sharp increase in the zone associated with the high amplitude synchronous oscillations, particularly for intermediate values of $\gamma$ (*Figure 2c*). In contrast, the transitions associated with manipulating $\gamma$ (particularly at high values of $\sigma$) led to an inverse U-shaped relationship: the network was relatively segregated at high and low levels of $\gamma$, but integrated at intermediate values of $\gamma$, albeit with a monotonic relationship when increasing $\sigma$ for low levels of $\gamma$ (*Figure 2d*). In addition, increases in between-hemisphere connectivity were more pronounced than within-hemisphere connectivity in the ordered state (within: 0.010 ± 0.017; between: 0.014 ± 0.013; $T_{2,848} = 7.104$; p=$10^{-12}$; see *Figure 2—figure supplement 4*). Together, these results suggest that neural gain and excitability act together to traverse a transition in network dynamics, maximizing inter-regional phase synchrony and integrating the functional connectome.

## Neural gain increases topological complexity and temporal variability

Having identified a relationship between neural gain and network architecture, we next investigated the putative topological benefit of this trade-off. A measure that characterizes the topological balance between integration and segregation is communicability (*Estrada and Hatano, 2008*), which quantifies the number of short paths that can be traversed between two regions of a network (*Mišić et al., 2015*). In networks with high communicability, individual regions are able to interact with a large proportion of the network through relatively short paths, which in turn may facilitate effective communication between otherwise segregated regions. In contrast to the relationship observed between neural gain and network integration, communicability was maximal at the critical boundaries between synchronous and asynchronous behavior (*Figure 3a–c*). Thus, the topological signature of the network was most effectively balanced between integration and segregation as the system transitioned between disorder and order through the modulation of inter-regional synchrony by subtle changes in neural gain.

Another important signature of complex systems is their flexibility over time. In previous work, we showed that the 'resting state' is characterized by significant fluctuations in network topology, in which the brain traverses between states that maximize either integration or segregation (*Shine et al., 2016a*). This variability was diminished during a cognitively challenging task, and the extent of integration was positively associated with improved task performance (*Shine et al., 2016a*). To determine whether these alterations in topological variability may have been related to changes in neural gain, we estimated the time-resolved mean participation coefficient ($B_T$) of the simulated BOLD time series and then determined whether the variability of this measure over time changed as a function of $\sigma$ and $\gamma$. We found that the variability of time-resolved integration within each trial was maximized across the critical boundary, as the network switched between disordered and ordered phase synchrony (*Figure 3d–f*). These results support the hypothesis that changes in

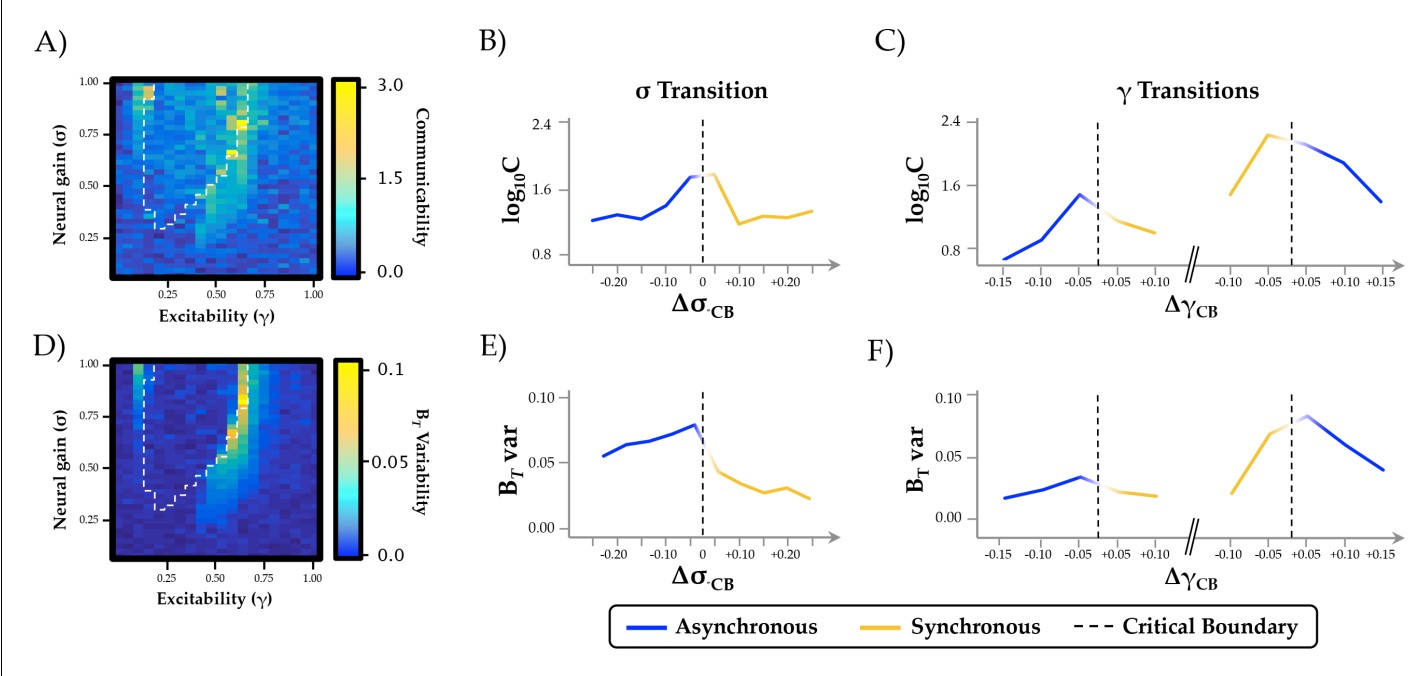

**Figure 3.** Topological and temporal relationships with phase regimen boundary. (**a-c**) network communicability was maximal following the σ boundary (Δσ$_{CP}$; mean across $0.2 \leq \gamma \leq 0.6$) and the immediately prior to the abrupt phase transition at high γ (Δγ$_{CP}$; mean across $0.3 \leq \sigma \leq 1.0$); (**d-f**) time-resolved between-module participation (B$_T$) was maximally variable with increasing σ and across the critical boundary at high γ.
DOI: https://doi.org/10.7554/eLife.31130.008

neural gain may control the temporal variability of network topology as a function of behavioral state.

## Gain-mediated integration is maximal in frontoparietal hub regions

To determine whether the influence of neural gain on network dynamics was related to the underlying structural connectivity of the brain, we estimated the 'rich club' architecture of the structural connectome (*Figure 4a*). Compared to low-degree nodes, rich club regions demonstrated an increase in 'realized' mean gain adjacent to the critical boundary (*Figure 4b*). In short, this means that activity within frontoparietal 'hub' regions (red in *Figure 4a*) was more strongly affected by the interaction between neural gain and network topology than in non-hub regions (blue/green in *Figure 4a*). Indeed, this result demonstrates that the 'realized' gain of individual regions is not simply related to the applied gain (i.e. input from the ascending noradrenergic system; (*Aston-Jones and Cohen, 2005*), but also non-linearly depends on afferent activity from topologically connected regions (*Figure 4c/d*). The observed effect was particularly evident for intermediate values of γ, suggesting that the hub regions were differentially impacted by neural gain at the critical boundary between the asynchronous and synchronous states. Interestingly, similar dissociations were observed when comparing regions with high and low diversity (*Figure 4—figure supplement 1*), suggesting a role for future experiments to disambiguate the importance of degree and diversity in the mediation of global network topology (*Bertolero et al., 2017*). However, given the substantial overlap between regions in the 'rich' and diverse' clubs (73% of regions were found in both groups), our results confirm a crucial role for frontoparietal regions in the control of network-level integration as a function of ascending neuromodulatory gain.

## Discussion

We used a combination of computational modeling and graph theoretical analyses, quantifying the relationship between ascending neuromodulation and network-level integration in order to test a direct prediction from a previous neuroimaging study (*Shine et al., 2016a*). We found that

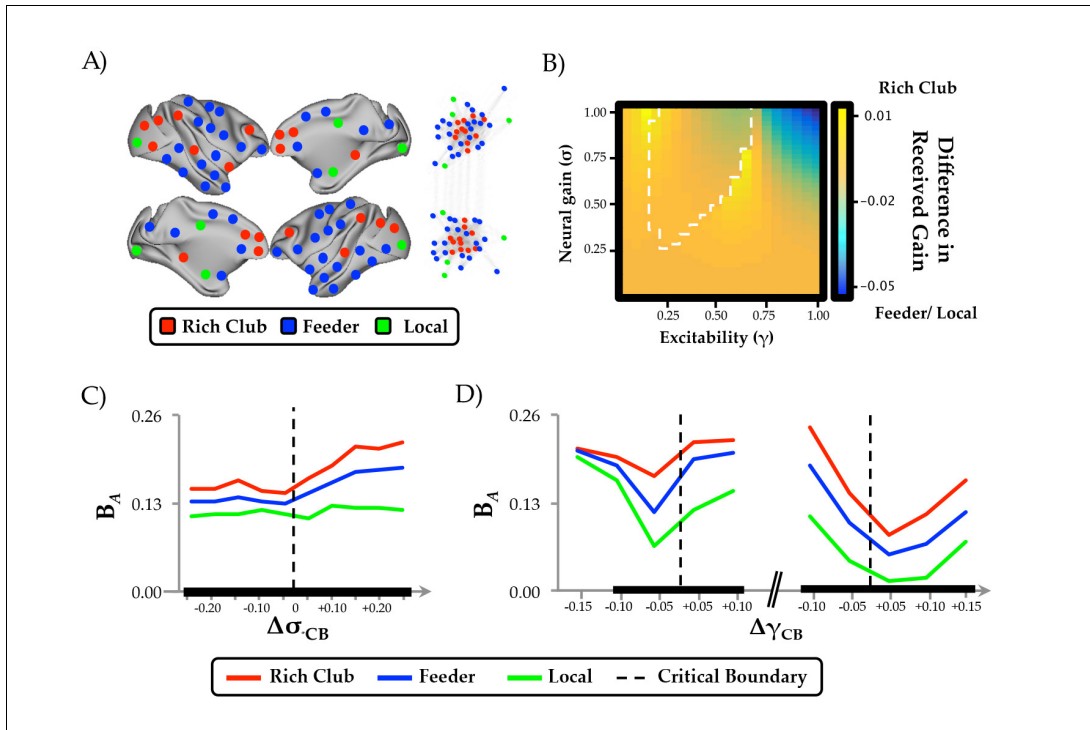

**Figure 4.** Regional clustering results. (**a**) regions from the CoCoMac data organized according to rich club (red), feeder (blue) or local (green) status, along with a force-directed plot of the top 10% of connections (aligned by hemisphere), colored according to structural hub connectivity status; (**b**) the rich club cluster demonstrated an increase in realized mean gain (the relative output as a function of its' unique topology) at the bifurcation boundary, compared to feeder and local nodes, which showed higher realized gain at high levels of σ and γ; (**c**) the three clusters of regions also demonstrated differential responses to neural gain; and (**d**) excitability. The black lines in (c) and (d) denote significant differences in B$_A$ between the two groups.

DOI: https://doi.org/10.7554/eLife.31130.009

The following figure supplements are available for figure 4:

**Figure supplement 1.** Diverse Club.
DOI: https://doi.org/10.7554/eLife.31130.010
**Figure supplement 2.** Clustering coefficient.
DOI: https://doi.org/10.7554/eLife.31130.011

increasing neural gain transitioned network dynamics across a bifurcation from disordered to ordered phase synchrony (*Figure 2b*) with a shift from a segregated to integrated neural architecture (*Figure 2e* and *Figure 2—figure supplement 1*). The critical boundary between these two states was associated with maximal communicability and temporal topological variability (*Figure 3*). Finally, the effect of neural gain was felt most prominently in high-degree frontoparietal network hubs (*Figure 4* and *Figure 4—figure supplement 2*). Together, these results confirm our prior hypothesis and complement an emerging view of the brain that highlights a mechanistic bridge between ascending arousal systems and cognition (*Shine et al., 2016a*), providing a potential mechanistic explanation for the long-standing notion that noradrenergic activity demonstrates an inverted U-shaped curve with cognitive performance (*Robbins and Arnsten, 2009*, *Figure 1a*).

The major result from our study is that network-level fluctuations between segregation and integration in functional (BOLD) networks reflect an underlying transition in synchrony of faster neuronal oscillations, thus providing a previously unknown link between temporal scales in the brain (*Figure 2b*). At low levels of γ and σ, the governing equations are strongly stable (damped), so that all excursions from equilibrium must be driven by local noise – that is, regions are relatively insensitive to incoming inputs (*Figure 1b/c*). As γ and σ increase, local activity approaches an instability, and consequently incoming activity is able to substantially influence activity in target regions. This

causes changes in the emergent whole-brain dynamics evident at both the short time scale of brain oscillations and the long time scale of BOLD correlation. A stark transition occurs at a critical point in the parameter space (denoted by the boundary between blue and yellow in *Figure 2b*), whereby small increases in σ lead to substantial alterations in the phase relationships between regions. Specifically, the network abruptly shifts from stable equilibrium to high-amplitude synchronized oscillation, facilitating an increase in effective communication between otherwise topologically distant regions (*Fries, 2005*; *Varela et al., 2001*). This same transition point is associated with a peak in informational complexity (*Figure 3*), further suggesting the importance of criticality in maximizing the information processing capacity of global network topology. Notably, the transition is also accompanied by a peak in the topological variability over time: hence a dynamic instability amongst fast neuronal oscillations yields increased network fluctuations at very slow time scales, again highlighting the crucial role of criticality to multi-scale neural phenomena (*Cocchi et al., 2017*).

The effect of neural gain on topology was greatest in a bilateral network of high-degree fronto-parietal cortical regions (*Figure 4*). This suggests that the recruitment of these hub regions at intermediate levels of excitability and neural gain shifts collective network dynamics across a bifurcation, increasing effective interactions between otherwise segregated regions. This result underlines the effective influence of the structural 'rich club' (*Figure 4*), which in addition to providing topological support to the structural connectome (*van den Heuvel and Sporns, 2013*), may also facilitate the transition between distinct topological states. This relationship has been demonstrated previously in other studies, either by manipulating the excitability parameter alone (*Deco et al., 2017*; *Zamora-López et al., 2016*), or through the alteration of the intrinsic dynamics of the 2d oscillator model (*Curto et al., 2009*; *Safaai et al., 2015*), thus providing a strong conceptual link between structural topology and emergent dynamics. Crucially, the integrated states facilitated by gain-mediated hub recruitment have been shown to underlie effective cognitive performance (*Shine et al., 2016a*), episodic memory retrieval (*Westphal et al., 2017*) and conscious awareness (*Barttfeld et al., 2015*; *Godwin et al., 2015*), confirming the importance of ascending neuromodulatory systems for a suite of higher-level behavioral capacities.

Overall, our findings broadly support the predictions of the neural gain hypothesis of noradrenergic function (*Aston-Jones and Cohen, 2005*). For instance, manipulating neural gain, a plausible instantiation of the effects of ascending noradrenergic tone in the brain (*Servan-Schreiber et al., 1990*), led to marked alterations in network topology. Given the demonstrated links between network topology and cognitive function (*Cohen and D'Esposito, 2016*; *Hearne et al., 2017*; *Shine et al., 2016a*; *Shine and Poldrack, 2017*), our work thus provides a plausible mechanistic account of the long-standing notion of a nonlinear relationship between catecholamine levels and effective cognitive performance (*Robbins and Arnsten, 2009*; *Shine et al., 2016a*; *Figure 1a*). However, it bears mention that our model highlighted a relationship between neural gain, excitability and network topology, in which there was an inverted-U shaped relationship observed between excitability and integration that was related to two separate bifurcations (*Figure 2—figure supplement 2*). In contrast, the effect of neural gain on topology was demonstrably more linear, particularly at intermediate levels of γ (*Figure 2*). Importantly, although noradrenaline has been directly linked to alterations in gain (*Servan-Schreiber et al., 1990*), there is also reason to believe that noradrenergic tone should have a demonstrable effect on excitability (*Curto et al., 2009*; *Safaai et al., 2015*; *Stringer et al., 2016*). Combined with our observation of the importance of the interaction between neural gain and high-degree (*Figure 4*), diverse (*Figure 4—figure supplement 1*) hub regions, our results thus represent an extension of the neural gain hypothesis that integrates the ascending arousal system with the constraints imposed by multiple order parameters and structural network topology.

In addition, our results also align with previous hypotheses that highlighted the importance of α2-adrenoreceptor mediated hub recruitment with increasing concentrations of noradrenaline, particularly in the frontal cortex (*Robbins and Arnsten, 2009*; *Sara, 2009*). However, our findings are inconsistent with the hypothesis that neural gain mediates an increase in tightly clustered patterns of neural interactions (*Eldar et al., 2013*). In contrast to this prediction, our simulations showed that measures that reflect an increase in local clustering, such as modularity and the mean clustering coefficient (*Figure 4—figure supplement 2*), did not increase as a function of neural gain in the same manner as other measures, such as the mean participation coefficient. Therefore, our results suggest that an increase in functional integration (and hence, a concomitant decrease in local

clustering) is a more effective indicator of the topological influence of increasing neural gain. However, it bears mention that the hypothesized relationship between clustering and neural gain was presented in the context of a focused learning paradigm (*Eldar et al., 2013*), whereas our data were not modeled in an explicit behavioral context. As such, future studies are required to disambiguate the relative relationship between neural gain and network topology as a function of task performance.

Prior computational studies have demonstrated a link between the structural and functional connectome, with the broad repertoire of functional network dynamics bounded by structural constraints imposed by the white-matter backbone of the brain (*Deco and Jirsa, 2012*; *Honey et al., 2007*, *2009*). While the targeted role of gain modulation on local neuronal dynamics have been studied (*Freeman, 1979*), the impact of gain on functional network organization has not been pursued. Here, we have demonstrated a putative mechanism by which a known biological system (namely, the ascending noradrenergic system) can mediate structural-functional changes, essentially by navigating the functional connectome across a topological landscape characterized by alterations in oscillatory synchrony. However, the direct relationship between neural gain manipulation and the ascending noradrenergic system is likely to represent an oversimplification. Indeed, given the complexity and hierarchical organization of the brain, it is almost certain that other functional systems, such as the thalamus (*Hwang et al., 2017*) and fast-spiking interneurons (*Stringer et al., 2016*), play significant roles in mediating neural gain and hence, the balance between integration and segregation. Further studies are required to interrogate these mechanisms more directly.

A somewhat surprising result of our simulation is the link between phase- and amplitude-related measures of neuronal coupling. It has been known for some time that the BOLD signal is insensitive to the relative phase of underlying neural dynamics (*Foster et al., 2016*), relating more closely to changes in the local oscillator frequency and fluctuations in the relative amplitude of neural firing. Indeed, each of the model parameters used in our experiment (i.e., gain and coupling) exerts a complex influence on both the oscillator frequencies (and hence, the BOLD activity) and the global synchrony (and hence, the BOLD correlations). Moreover, in coupled oscillator systems such as this, the order parameter acts as a 'mean field' that feeds back and influences local dynamics (see e.g. *Breakspear et al., 2010*). Based on this knowledge, we can infer that estimates of connectivity using BOLD time series relate to covariance in amplitude fluctuations among pairs of regions, rather than alterations in phase synchrony. This clarification is important for modern theories of functional neuroscience, as synchronous relationships between regions in the phase domain have been used to explain effective communication between neural regions (*Fries, 2015*; *Lisman and Jensen, 2013*; *Siegel et al., 2009*), in which the precise timing between spiking populations determines the efficacy of information processing. Our results suggest a surprisingly robust link between these two measures, such that an integrated network with increased inter-modular amplitude correlation coincides with a peak in ordered phase synchrony between regions. In our model, the peak of network variability occurs at the critical transition between disordered and ordered phases, where the local dynamic states shows the most variability and where fast stochastic perturbations are most able to influence slow amplitude fluctuations. However, while our model provides evidence linking neural gain to functional integration, advanced models that display a broader variety of non-linear dynamics (*Breakspear, 2017*) are required to test these hypotheses more directly.

Together, our results suggest that the balance between integration and segregation relates to alterations in neural gain that exist within a 'zone' of maximal communicability and temporal variability. Our findings thus highlight important constraints on contemporary models of brain function, while also providing crucial implications for understanding effective brain function during task performance or as a function of neurodegenerative or psychiatric disease.

## Materials and methods

### Dynamical network modeling

The Virtual Brain software (*Sanz Leon et al., 2013*) was used to simulate neural activity across a lattice of parameter points in which we manipulated the inter-regional coupling between regions using both a gain parameter and an excitability parameter. Specifically, we used a generic 2-dimensional oscillator model (*Equations 1 and 2*) to create time series data that represents neural activity via

two variables (the membrane potential and a slow recovery variable). This equation is based upon a modal approximation (*Stefanescu and Jirsa, 2008*) of a population of Fitzhugh-Nagumo neurons (*Izhikevich and FitzHugh, 2006*). The neuronal dynamics are given by,

$$\dot{V}_i(t) = 20\Big(W_i(t) + 3V_i(t)^2 - V_i(t)^3 + \gamma I_i\Big) + \xi_i(t), \tag{1}$$

$$\dot{W}_i(t) = 20(-W_i(t) - 10V_i(t)) + \eta_i(t), \tag{2}$$

where $V_i$ represents the local mean membrane potential and $W_i$ represents the corresponding slow recovery variable at node $i$. Stochastic fluctuations are introduced additively through the white noise processes $\eta_i$ and $\xi_i$, drawn independently from Gaussian distributions with zero mean and unit variance. The synaptic current $I_i$ arise from time-delayed input from other regions modulated in strength by the global excitability parameter $\gamma$. This input arises after the mean membrane potential $V$ in distant nodes is converted into a firing rate via a sigmoid-shaped activation function $S$, and then transmitted with axonal time delays through the connectivity matrix. Hence the synaptic current at node $i$ is given by,

$$I_i = \sum_j A_{ij}\, S_j\big(t - \tau_{ij}\big) \tag{3}$$

where $A_{ij}$ is the directed connectivity matrix derived from the 76 region CoCoMac connectome (*Kötter, 2004*), and $\tau_{ij}$ is the corresponding time delay computed from the length of fiber tracts estimated by diffusion spectrum imaging (*Sanz Leon et al., 2013*). The conversion from regional membrane potential to firing rate is given by a sigmoid-shaped activation function,

$$S_i(t) = \frac{1}{1 + e^{-\sigma(V_i(t) - m)}}, \tag{4}$$

where $\sigma$ is the (global) gain parameter and the sigmoid activation function is shifted to center at $m$. These equations were integrated using a stochastic Heun method (*Rüemelin, 1982*).

The simulated neuronal data were fed through a Balloon-Windkessel model to simulate realistic Blood Oxygen Level Dependent signals (*Friston et al., 2000*). The simulated BOLD time series were band-pass filtered (0.01–0.1 Hz) and the Pearson's correlation was then computed (and normalized using Fisher's r-to-Z transformation).

We manipulated the inter-regional neural gain parameter $\sigma$ and the regional excitability $\gamma$ through a range of values (between 0–1). After aligning the sensitive region of the sigmoid function with its mean input ($m$ = 1.5). Consistent with the effects of relatively diffuse projections from the locus coeruleus to cortex, all regions were given the same values of the $\sigma$ and $\gamma$ parameter for each trial. All code is freely available at https://github.com/macshine/gain_topology (*Shine, 2018*). A copy is archived at https://github.com/elifesciences-publications/gain_topology.

## Integration and segregation

The Louvain modularity algorithm from the Brain Connectivity Toolbox (*Rubinov and Sporns, 2010*) was used to estimate time-averaged community structure. The Louvain algorithm iteratively maximizes the modularity statistic, $Q$, for different community assignments until the maximum possible score of $Q$ has been obtained (*Equation 5*). The modularity estimate for a given network is therefore a quantification of the extent to which the network may be subdivided into communities with stronger within-module than between-module connections. Here, we used the $Q$ parameter to estimate the extent of segregation within each graph,

$$Q = \frac{1}{v^+}\sum_{ij}\Big(w_{ij}^+ - e_{ij}^+\Big)\delta_{M_iM_j} - \frac{1}{v^+ + v^-}\sum_{ij}\Big(w_{ij}^- - e_{ij}^-\Big)\delta_{M_iM_j} \tag{5}$$

where $v$ is the total weight of the network (sum of all negative and positive connections), $w_{ij}$ is the weighted and signed connection between regions $i$ and $j$, $e_{ij}$ is the strength of a connection divided by the total weight of the network, and $\delta_{M_iM_j}$ is set to one when regions are in the same community and 0 otherwise. '+' and '–' superscripts denote all positive and negative connections, respectively. Consistent with previous work (*Eldar et al., 2013*), the mean clustering coefficient, which reflects the

proportion of closed 'triangles' in the binarized graph, was also used as a measure of segregation (**Rubinov and Sporns, 2010**).

For each level of neural gain, the community assignment for each region was assessed 100 times and a consensus partition was identified using a fine-tuning algorithm from the Brain Connectivity Toolbox (http://www.brain-connectivity-toolbox.net/). All graph theoretical measures were calculated on weighted and signed connectivity matrices (**Rubinov and Sporns, 2010**), and weak connections were retained using a consistency thresholding technique that identifies weak, yet consistent connections by identifying edges with minimal variance across multiple iterations (**Roberts et al., 2017**). In order to assess global, large-scale communities, the resolution parameter was set to 1.0 (higher values tune the algorithm to detect smaller communities, which instead reflect local, rather than global, clustering). This parameter was chosen by calculating the resolution value which maximized the Surprise (**Aldecoa and Marín, 2013**) between the community structure of the network at each level of gain and resolution and a random network defined using a cumulative hypergeometric distribution (see [**Aldecoa and Marín, 2013**]).

The participation coefficient, $B_A$ (**Equation 6**) quantifies the extent to which a region connects across all modules (i.e. between-module strength). As such, the mean participation coefficient can be used to estimate the extent of integration within a graph. The participation coefficient, $B_{Ai}$, for a given region $i$ is,

$$B_{Ai} = 1 - \sum_{s=1}^{n_M} \left( \frac{\kappa_{is}}{\kappa_i} \right)^2 \tag{6}$$

where $\kappa_{is}$ is the strength of the positive connections of region $i$ to regions in module $s$, and $\kappa_i$ is the sum of strengths of all positive connections of region $i$. The participation coefficient of a region is therefore close to one if its connections are uniformly distributed among all the modules and 0 if all of its links are within its own module. Finally, the global efficiency (mean inverse characteristic path length) and inverse modularity ($Q^{-1}$) were estimated for each element of the parameter space as adjunct measures of integration.

## Phase synchrony order parameter

To estimate the degree of phase synchrony at different points in the parameter space, we extracted the raw signal ($Vi$) from each region in the simulation and subtracted the least squares linear trend from each channel. We then computed the phase of the analytic signal for each channel using the Hilbert transform and then estimated the phase synchrony order parameter (across all channels), OP, which is given by,

$$\rho = \left| \frac{1}{N} \sum_{j=1}^{N} e^{i\theta_j} \right| \tag{7}$$

where $i = \sqrt{-1}$ and $\theta_j$ represents the oscillation phase of the $j$th region. Large values of $\rho$ denote phase alignment between regions (**Breakspear et al., 2010**; **Kuramoto, 1984**). The value of $\rho$ for each parameter combination was subsequently averaged over time and across sessions. By designating each parameter combination as resulting in either a synchronized ($\rho \geq 0.5$) or unsynchronized ($\rho < 0.5$) regime, we were able to determine whether network topology changes as a function of neural gain and excitability estimated from BOLD data coincided with changes of underlying phase synchrony. Specifically, we then separately grouped topological variables and within- and between-hemisphere connectivity according to their underlying $\rho$ value and then estimated an independent-samples t-test between the two groups. The standard deviation of the order parameter, $\rho$, was also calculated and averaged across sessions. Finally, the dwell times for regional fluctuations were estimated for a number of characteristic parameter choices and analyzed for evidence of Pareto (i.e. power law) scaling.

## Communicability

The communicability, $C$, between a pair of nodes $i$ and $j$ is defined as a weighted sum of the number of all walks connecting the pair of nodes (within weighted connectivity matrix, $A$) and has been shown to be equivalent to the matrix exponent of a binarized graph, $e^A$ (**Estrada and Hatano,**

*2008*). For ease of interpretation, we calculated the $\log_{10}$-transformed mean of communicability for each graph across iterations and values of neural gain.

$$C_{ij} = \sum_{k=0}^{\infty} \frac{(A^k)_{ij}}{k!} = e^A \tag{8}$$

## Topological variability

To estimate time-resolved functional connectivity between the 76 nodal pairs, we used a recently described statistical technique (Multiplication of Temporal Derivatives; (*Shine et al., 2015*); http://github.com/macshine/coupling), which is computed by calculating the point-wise product of temporal derivative of pairwise time series (*Equation 7*). To reduce the contamination of high-frequency noise in the time-resolved connectivity data, $M_{ij}$ was averaged over a temporal window ($w$ = 15 time points). Individual functional connectivity matrices were calculated within each temporal window, thus generating an unthresholded (signed and weighted) 3D adjacency matrix (region × region × time) for each participant. These matrices were then subjected to time-resolved topological analyses, which allowed us to estimate the participation coefficient for each region over time ($B_T$). We used the mean regional standard deviation of this measure to estimate time-resolved topological variability in the simulated data.

$$M_{ijt} = \frac{1}{w} \sum_{t}^{t+w} \frac{(dt_{it} \times dt_{jt})}{(\sigma_{dt_i} \times \sigma_{dt_j})} \tag{9}$$

for each time point, $t$, $M_{ij}$ is defined according to *Equation 1*, where $dt$ is the first temporal derivative of the $i^{th}$ or $j^{th}$ time series at time $t$, $\sigma$ is the standard deviation of the temporal derivative time series for region $i$ or $j$ and $w$ is the window length of the simple moving average. This equation can then be calculated over the course of a time series to obtain an estimate of time-resolved connectivity between pairs of regions.

## Structural rich club

To test whether changes associated with neural gain were mediated by highly-interconnected high-degree hubs, we identified a set of 'rich club' regions using the structural white matter connectome from the CoCoMac database (*Kötter, 2004*). Briefly, the degree of each node $i$ in the network was determined by calculating the number of links that node $i$ shared with $k$ other nodes in the network. All nodes that showed a number of connections of $\leq k$ were removed from the network. For the remaining network, the rich-club coefficient ($\Phi_k$) was computed as the ratio of connections present between the remaining nodes and the total number of possible connections that would be present when the set would be fully connected. We then normalized $\Phi_k$ relative to a set of random networks with similar density and connectivity distributions. When $\Phi_Z$ is greater than 1, the network can be said to display a 'rich club' architecture. Individual regions that are interconnected at the value of $k$ at which the network demonstrates a 'rich club' architecture are thus designated as 'rich club' nodes (n = 22). Any nodes outside of this group but still sharing a connection are labeled as 'feeder' nodes (n = 44), and regions disconnected from the rich club are designated as 'local' nodes (n = 10). The results were projected onto a standard surface representation of the macaque cortex (*Figure 4*). After segmenting the network in this fashion, we were able to estimate the realized mean gain and $B_A$ across the parameter space for regions according to their structural topology.

## Realized neural gain

While the neural gain parameter σ controls the *maximum* gain in each region within the simulation by setting the maximum slope of the sigmoid, the realized gain (mean ratio of sigmoid output to input) for each brain region depends upon the distribution of its input, and is greater when the input level is concentrated near the center of the sigmoid. We estimated the regional variation in effective or 'realized' neural gain by calculating the integral of the instantaneous sigmoid slope over its complete input range, weighted by the probability of each input level. We then compared these values as a function of nodal class (rich club vs other nodes) at each aspect of the parameter space.

## Reliability

We ran a number of subsequent tests to ensure that any observed changes in network topology were robust to the processing steps utilized in the analysis. Firstly, we re-analyzed data across a range of network thresholds (1–20%) and observed robust results (i.e. r > 0.75) for Q, mean $B_A$, mean communicability and the standard deviation of $B_T$ on graphs estimated between the 9–20% threshold range. Secondly, as the number of modules estimated from graphs can change as a function of network topology, we re-examined the topological characteristics of networks that were matched for the number of modules (N = 4) and found no significant differences to the topological signatures estimated on the whole group.

## Acknowledgements

We thank the creators of the Virtual Brain for their open-source software, Peter Bell for helpful comments, Bratislav Misic for sharing code and Joke Durnez for statistical advice.

## Additional information

### Funding

| Funder | Grant reference number | Author |
| --- | --- | --- |
| National Health and Medical Research Council | GNT1072403 | James M Shine |

The funders had no role in study design, data collection and interpretation, or the decision to submit the work for publication.

### Author contributions

James M Shine, Conceptualization, Data curation, Formal analysis, Funding acquisition, Validation, Investigation, Visualization, Writing—original draft, Project administration, Writing—review and editing; Matthew J Aburn, Resources, Data curation, Software, Formal analysis, Investigation, Visualization, Methodology, Writing—review and editing; Michael Breakspear, Formal analysis, Supervision, Methodology, Writing—review and editing; Russell A Poldrack, Conceptualization, Formal analysis, Supervision, Methodology, Writing—review and editing

### Author ORCIDs

James M Shine http://orcid.org/0000-0003-1762-5499
Russell A Poldrack http://orcid.org/0000-0001-6755-0259

### Decision letter and Author response

Decision letter https://doi.org/10.7554/eLife.31130.016
Author response https://doi.org/10.7554/eLife.31130.017

## Additional files

### Supplementary files

• Transparent reporting form
DOI: https://doi.org/10.7554/eLife.31130.012

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
