## [Decision Letter]

Thank you for submitting your article "The modulation of neural gain facilitates a transition between functional segregation and integration in the brain" for consideration by *eLife*. Your article has been favorably evaluated by a Senior Editor and three reviewers, one of whom, Gustavo Deco (Reviewer #1), is a member of our Board of Reviewing Editors. The following individuals involved in review of your submission have agreed to reveal their identity: Tobias H Donner (Reviewer #2); Maxwell Bertolero (Reviewer #3).

The reviewers have discussed the reviews with one another and the Reviewing Editor has drafted this decision to help you prepare a revised submission.

Your paper presents a very useful integration a modeling framework and graph theoretical analysis, providing a mechanistic explanation of how neuromodulation balances the equilibrium between segregation and integration of functional information. All reviewers found the paper to be well written and the results to be relevant for the broader neuroscience community. All reviewers support the publication of the paper in *eLife*, provided that you address the following issues in your revision.

1) Additional analyses:

1a) The participation coefficient is a great measure of integration, but it needs to be supplemented with other measures. You should test the integration hypotheses with the participation coefficient, Q (inverse), and efficiency, and more carefully lay out why these measures make sense for testing the hypotheses regarding integration and segregation.

Moreover, the participation coefficient is a nodal measure, but you only report the mean – are all nodes equally increasing their participation coefficients? Are low or high nodes increasing their participation coefficients? These analyses will give more insight into exactly what is occurring in the network as a result of neural gain.

Rationale behind this request: The participation coefficient is a relative measure of the diversity a node's connections across communities. Thus, consider a node with strong connectivity to community A and weak connectivity to communities B-F. If the node simply decreases connectivity to community A, its participation coefficient increases. While this is likely rare, it can occur, and this might not capture the intuition of increased integration. On the other hand, perhaps it does, as the node is interacting with communities in a more equal fashion. Moreover, if other nodes from community A shift to community B, the participation coefficient of the node will increase, even if none of its connections did. Because of these ambiguities, supplementing the mean participation coefficient with Q and efficiency (the inverse of the sum of shortest paths) would be ideal. While Q is used as a measure of segregation, which is appropriate, the reasoning behind this is not laid out. It should be.

1b) You show that the profile for global phase synchrony (computed from the regional membrane potentials) is very similar to the profiles for the graph-theoretical measures that are based on the correlations of (much slower) local BOLD time series (Figure 2). This observation is non-trivial. It should be complemented by an analysis of a measure of the dynamics of global phase synchrony – for instance, the standard deviation of global phase synchrony, which has been used to quantify metastability. Does this measure exhibit the same profile as those in Figure 2? Along that line, please discuss *why* the similarity between BOLD correlation topology and global phase synchrony emerges. Do you have an intuition for this?

1c) The "Gain mediated integration is maximal in frontoparietal hub regions" findings should be complemented by analyzing the "diverse club" or the regions with high participation coefficients (Bertolero et al., 2017), which was found to be more highly interconnected than the rich club in the macaque.

2) Discussion, embedding of the present results into the existing literature.

2a) The Discussion should elaborate on the facts that (i) several of the results are motivated and expected from previous work, and (ii) the excitability parameter in your model is analogous to the global coupling, which has been used as control parameter in previous studies. The main new contribution here seems to be the addition of a second control parameter, the global gain.

Please note that, all reviewers appreciate the need of implementing the effects of neuromodulation effect explicitly, but they feel that the readership would benefit from a more balanced discussion of the context.

Rationale behind this request: Several previous studies have shown that networks of oscillators reach a balance between integration and segregation, a maximal complexity, and the largest variability of temporal networks at the critical point separating the asynchronous and synchronous phases (e.g., Schmidt et al., 2015; Zamora-Lopez et al. 2016; Deco et al. 2017). These studies also showed that rich clubs have a leading role concerning integration/segregation and complexity. The network can be displaced from one phase to the other by changing a control parameter, which is usually chosen to be the global coupling or connectivity strength – analogous to the excitability parameter used in your model. In addition, you manipulate a second control parameter, namely the slope of the transfer function that converts inputs into firing rates. Increasing the slope brings the system from asynchronous state to the synchronous state, and, as shown in Equation [3], has the same effect than a global coupling.

2b) Changes in both parameters (gain and excitability) are necessary for producing the effects of interest in your model. In particular, the graph-theoretical measures show a non-monotonic (inverted U) dependency on excitability, but not on gain. Still, the title and much of your Discussion focusses on gain only, drawing firm links to the Astone-Jones and Cohen Adaptive Gain Theory. Please discuss more explicitly the interesting effects of excitability. Specifically: How can you reconcile the predominance of inverted U patterns for excitability but not gain, with the Aston-Jones and Cohen framework.

2c) Previous work by the labs of Ken Harris and Stefano Panzeri has also used the FitzHugh-Nagumo model to characterise the effects of state/neuromodulation on cortical dynamics – but by modulating the dynamics of the 2D-oscillator itself (Curto et al., 2009; Safaai et al., PNAS, 2015). This work should be discussed and related to your current approach.

3) Clarifications

3a) In general, all reviewers appreciate that the paper was written for a general audience. However, please include all equations and note in the main text where they exist. For example, please include the equation for Communicability and expand the explanation – is it all walks, or all walks that are part of all shortest walks between all nodes? It is not clear in the Results or Materials and methods section.

3b) Please unpack the rationale behind using the correspondence between structural and functional connectivity as criterion for delineating the "plausible" sub-space of parameter combinations. This is important because work into neuromodulation in small circuits shows that neuromodulation can reconfigure circuits, thus overriding the structural connectome (Marder, Neuron, 2012).

3c) It seems that Vi is used for the computation of phase synchrony (i.e., not the neural activity passed through the hemodynamics). This should be made explicit in the Materials and methods. (Terms like "raw signal" or "neural data" are ambiguous.)

3d) Figure 2 is described as inverted U relationship, a description that glosses over the left flank of this plot. The relationship clearly is monotonic, but not an inverted U.

3e) Figure 2—figure supplement 3 is missing axis labels and won't be understandable for a broad audience. Please explain what it shows and what that means.

---

## [Author Response]

[…] All reviewers support the publication of the paper in eLife, provided that you address the following issues in your revision.1) Additional analyses:1a) The participation coefficient is a great measure of integration, but it needs to be supplemented with other measures. You should test the integration hypotheses with the participation coefficient, Q (inverse), and efficiency, and more carefully lay out why these measures make sense for testing the hypotheses regarding integration and segregation.

We thank the reviewers for their thoughtful comments regarding the potential limitations associated with the interpretation of the participation coefficient as a unitary measure to estimate topological integration. As requested, we have calculated the inverse of the modularity statistic (Q^-1^), taking care to iterate the algorithm a number of times due to stochasticity, and the global efficiency (GE) for the time-averaged connectivity matrix associated with each element of the parameter space. The expected result was apparent for both variables – i.e., increased Q^-1^ and GE as a function of increasing α and intermediate γ. We have added these two results to the manuscript, and included a separate figure, Figure 2—figure supplement 1.

The following text was added to the manuscript:

“Similar patterns were observed for other topological measures of integration, such as the inverse modularity (Q^-1^) and global efficiency (Figure 2—figure supplement 1).”

Moreover, the participation coefficient is a nodal measure, but you only report the mean – are all nodes equally increasing their participation coefficients? Are low or high nodes increasing their participation coefficients? These analyses will give more insight into exactly what is occurring in the network as a result of neural gain.

We thank the reviewers for this comment, and have conducted an adjunct analysis (similar to the experiment conducted for the ‘rich club’ regions) for the so-called ‘diverse club’ – i.e., regions with elevated participation. Specifically, we calculated the mean B*_A_* across the parameter space and then identified the top 30% of nodes (to match the number of identified rich club regions) – 16/22 regions (~73%) were present in both the ‘diverse’ and ‘rich’ club. We then estimated the mean B*_A_* across the parameter space for regions within the ‘diverse club’ and contrasted their gain- and excitability-mediated alterations in B*_A_*. The results (presented in Figure 4—figure supplement 1) were comparable to those observed in the rich club analysis, suggesting a need for future experiments to delineate the precise topological role of degree and diversity in mediating the functional signature of brain activity over time.

The following text was added to the manuscript:

“Interestingly, similar dissociations were observed when comparing regions with high and low diversity (Figure 4—figure supplement 1), suggesting a role for future experiments to disambiguate the importance of degree and diversity in the mediation of global network topology (Bertolero et al., 2017). However, given the substantial overlap between regions in the ‘rich’ and diverse’ clubs (73% of regions were found in both groups), our results confirm a crucial role for frontoparietal regions in the control of network-level integration as a function of ascending neuromodulatory gain.”

Rationale behind this request: The participation coefficient is a relative measure of the diversity a node's connections across communities. Thus, consider a node with strong connectivity to community A and weak connectivity to communities B-F. If the node simply decreases connectivity to community A, its participation coefficient increases. While this is likely rare, it can occur, and this might not capture the intuition of increased integration. On the other hand, perhaps it does, as the node is interacting with communities in a more equal fashion. Moreover, if other nodes from community A shift to community B, the participation coefficient of the node will increase, even if none of its connections did. Because of these ambiguities, supplementing the mean participation coefficient with Q and efficiency (the inverse of the sum of shortest paths) would be ideal. While Q is used as a measure of segregation, which is appropriate, the reasoning behind this is not laid out. It should be.

We thank the reviewers for their thoughtful comments and have added a more explicit justification of the utilization of the participation coefficient and modularity statistic as indices of integration and segregation, respectively.

The following text was added to the manuscript:

“This allowed us to estimate the amount of integration in the time-averaged functional connectivity matrix across the parameter space (Figure 2[…] The converse situation (i.e., segregation) can thus be indexed by low mean B_A_ scores, or alternatively by the modularity statistic, Q.”

1b) You show that the profile for global phase synchrony (computed from the regional membrane potentials) is very similar to the profiles for the graph-theoretical measures that are based on the correlations of (much slower) local BOLD time series (Figure 2). This observation is non-trivial. It should be complemented by an analysis of a measure of the dynamics of global phase synchrony – for instance, the standard deviation of global phase synchrony, which has been used to quantify metastability. Does this measure exhibit the same profile as those in Figure 2?

We appreciate the constructive suggestion to better quantify the nature of the dynamics at the critical boundary and, as suggested, have now estimated the standard deviation of the order parameter as a function of neural gain and excitability.

The following text was added to the manuscript:

“Across the parameter space, we observed two clear states (Figure 2): one associated with high (ρ ≥ 0.5; yellow) and one with low (ρ < 0.5; blue) mean synchrony, with a clear critical boundary demarcating the two states (dotted white line in Figure 2/B) that was associated with a relative increase in the standard deviation of the order parameter (Figure 2—figure supplement 2).”

We next undertook additional analyses to better quantify the nature of the dynamics at the critical boundary. We first observed that the standard deviation of the order parameter increased around the boundary delineating the low and high ordered states (Figure 2—figure supplement 2). This increase in fluctuation amplitude suggests the presence of complex dynamics, such as criticality, metastability or multistability (see Cocchi et al., 2017 and Heitman and Breakspear, 2017 for further discussion). To disambiguate these possibilities, we next studied the probability distribution of the fluctuations in the order parameter (Figure 2—figure supplement 2 and Roberts et al., 2015). Close to the boundary, we observed a truncated Pareto (power law) scaling regime spanning up to two orders of magnitude (Figure Y2—figure supplement 2B), consistent with a critical bifurcation in a complex, many-bodied system. After crossing the boundary, this relationship develops a ‘knee’ above the power-law scaling (Figure 2—figure supplement 2), consistent with the emergence of a characteristic temporal scale in a super-critical system (Roberts et al. 2015). These observations suggest that the system undergoes a bifurcation across a critical boundary as the synchronization manifold loses stability.

The following text was added to the manuscript:

“To further disambiguate the system-level dynamics, we studied the probability distribution of the fluctuations in the order parameter. […] These observations suggest that the system undergoes a bifurcation across a critical boundary as the synchronization manifold loses stability.”

The following text was also added to the manuscript:

“The standard deviation of the order parameter, ρ, was also calculated and averaged across sessions. Finally, the dwell times for regional fluctuations were estimated for a number of characteristic parameter choices and analyzed for evidence of Pareto (i.e. power law) scaling.”

Along that line, please discuss why the similarity between BOLD correlation topology and global phase synchrony emerges. Do you have an intuition for this?

The factors mediating the relationship between the BOLD correlation topology and the global phase synchrony are quite complex. Recall that in the present model, the BOLD is simulated by driving the synaptic current through a Balloon-Windkessel model. Therefore, an increase in local neuronal activity (such as by an increase in the local oscillator frequency) will drive an increase in the local BOLD amplitude. Likewise, an increase in inter-system synchrony increases the order parameter. Therefore, to mediate the observed relationship between the BOLD topology and the global phase synchrony, there must exist a relationship between the global phase synchrony and the mean frequency of all oscillators: this is indeed the case (*r* = 0.21, *p* = 2.2x10^-09^) (Author response image 1)

Moreover, this relationship is mediated in a complex manner (through the neuronal state equation) by dependence of the global synchrony on both coupling (γ) and gain (σ) (Author response image 2).

**Author response image 2. respfig2:** 

In sum, both model parameters (gain and coupling) exert a complex influence on both the oscillator frequencies (and hence, the BOLD activity) and the global synchrony (and hence, the BOLD correlations). Moreover, in coupled oscillator systems such as this, the order parameter acts as a “mean field” that feeds back and influences local dynamics (see e.g. Breakspear et al., 2010).

The following text was added to the manuscript:

“A somewhat surprising result of our simulation is the link between phase- and amplitude-related measures of neuronal coupling. […] Moreover, in coupled oscillator systems such as this, the order parameter acts as a “mean field” that feeds back and influences local dynamics (see e.g. Breakspear et al., 2010).”

1c) The "Gain mediated integration is maximal in frontoparietal hub regions" findings should be complemented by analyzing the "diverse club" or the regions with high participation coefficients (Bertolero et al., 2017), which was found to be more highly interconnected than the rich club in the macaque.

We agree that the relationship between ascending neural gain and the diversity of inter-regional connectivity is likely to be important for global integration. As such, we have conducted an adjunct analysis (similar to the experiment conducted for the ‘rich club’ regions) for the so-called ‘diverse club’ – i.e., regions with elevated participation. Specifically, we calculated the mean B*_A_* across the parameter space and then identified the top 30% of nodes (to match the number of identified rich club regions) – 16/22 regions (~73%) were present in both the ‘diverse’ and ‘rich’ club. We then estimated the mean B*_A_* across the parameter space for regions within the ‘diverse club’ and contrasted their gain- and excitability-mediated alterations in B*_A_*. The results (presented in Figure 4—figure supplement 1) were comparable to those observed in the rich club analysis, suggesting a need for future experiments to delineate the precise topological role of degree and diversity in mediating the functional signature of brain activity over time.

The following text was added to the manuscript:

“Interestingly, similar dissociations were observed when comparing regions with high and low diversity (Figure 4—figure supplement 1), suggesting a role for future experiments to disambiguate the importance of degree and diversity in the mediation of global network topology (Bertolero et al., 2017). However, given the substantial overlap between regions in the ‘rich’ and diverse’ clubs (73% of regions were found in both groups), our results confirm a crucial role for frontoparietal regions in the control of network-level integration as a function of ascending neuromodulatory gain.”

2) Discussion, embedding of the present results into the existing literature.2a) The Discussion should elaborate on the facts that (i) several of the results are motivated and expected from previous work, and (ii) the excitability parameter in your model is analogous to the global coupling, which has been used as control parameter in previous studies. The main new contribution here seems to be the addition of a second control parameter, the global gain.Please note that, all reviewers appreciate the need of implementing the effects of neuromodulation effect explicitly, but they feel that the readership would benefit from a more balanced discussion of the context.

We thank the reviewers for their comments, and have added a discussion of this work to the manuscript.

The following text was added to the manuscript:

“The effect of neural gain on topology was greatest in a bilateral network of high-degree frontoparietal cortical regions (Figure 4). […] Crucially, the integrated states facilitated by gain-mediated hub recruitment have been shown to underlie effective cognitive performance (Shine et al., 2016a), episodic memory retrieval (Westphal et al., 2017) and conscious awareness (Barttfeld et al., 2015; Godwin et al., 2015), confirming the importance of ascending neuromodulatory systems for a suite of higher-level behavioral capacities.”

Rationale behind this request: Several previous studies have shown that networks of oscillators reach a balance between integration and segregation, a maximal complexity, and the largest variability of temporal networks at the critical point separating the asynchronous and synchronous phases (e.g., Schmidt et al., 2015; Zamora-Lopez et al. 2016; Deco et al. 2017). These studies also showed that rich clubs have a leading role concerning integration/segregation and complexity. The network can be displaced from one phase to the other by changing a control parameter, which is usually chosen to be the global coupling or connectivity strength – analogous to the excitability parameter used in your model. In addition, you manipulate a second control parameter, namely the slope of the transfer function that converts inputs into firing rates. Increasing the slope brings the system from asynchronous state to the synchronous state, and, as shown in Equation [3], has the same effect than a global coupling.

We thank the reviewers for their comments, and have added a discussion of this work to the manuscript.

The following text was added to the manuscript:

“This relationship has been demonstrated previously in other studies, either by manipulating the excitability parameter alone (Zamora-López et al., 2016) (Deco et al., 2017), or through the alteration of the intrinsic dynamics of the 2d oscillator model (Curto et al., 2009; Safaai et al., 2015), thus providing a strong conceptual link between structural topology and emergent dynamics. Crucially, the integrated states facilitated by gain-mediated hub recruitment have been shown to underlie effective cognitive performance (Shine et al., 2016a), episodic memory retrieval (Westphal et al., 2017) and conscious awareness (Barttfeld et al., 2015; Godwin et al., 2015), confirming the importance of ascending neuromodulatory systems for a suite of higher-level behavioral capacities.”

2b) Changes in both parameters (gain and excitability) are necessary for producing the effects of interest in your model. In particular, the graph-theoretical measures show a non-monotonic (inverted U) dependency on excitability, but not on gain. Still, the title and much of your Discussion focusses on gain only, drawing firm links to the Astone-Jones and Cohen Adaptive Gain Theory. Please discuss more explicitly the interesting effects of excitability. Specifically: How can you reconcile the predominance of inverted U patterns for excitability but not gain, with the Aston-Jones and Cohen framework.

We thank the reviewers for their comments, and have added a discussion of this work to the manuscript.

The following text was added to the manuscript:

*“*Overall, our findings broadly support the predictions of the neural gain hypothesis of noradrenergic function (Aston-Jones and Cohen, 2005b). […] Combined with our observation of the importance of the interaction between neural gain and high-degree (Figure 4), diverse (Figure 4—figure supplement 1) hub regions, our results thus represent an extension of the neural gain hypothesis that integrates the ascending arousal system with the constraints imposed by multiple order parameters and structural network topology.”

2c) Previous work by the labs of Ken Harris and Stefano Panzeri has also used the FitzHugh-Nagumo model to characterise the effects of state/neuromodulation on cortical dynamics – but by modulating the dynamics of the 2D-oscillator itself (Curto et al., 2009; Safaai et al., PNAS, 2015). This work should be discussed and related to your current approach.

We thank the reviewers for their comments, and have added a discussion of this work to the manuscript.

The following text was added to the manuscript:

“This relationship has been demonstrated previously in other studies, either by manipulating the excitability parameter alone (Zamora-López et al., 2016) (Deco et al., 2017), or through the alteration of the intrinsic dynamics of the 2d oscillator model (Curto et al., 2009; Safaai et al., 2015), thus providing a strong conceptual link between structural topology and emergent dynamics. Crucially, the integrated states facilitated by gain-mediated hub recruitment have been shown to underlie effective cognitive performance (Shine et al., 2016a), episodic memory retrieval (Westphal et al., 2017) and conscious awareness (Barttfeld et al., 2015; Godwin et al., 2015), confirming the importance of ascending neuromodulatory systems for a suite of higher-level behavioral capacities.”

3) Clarifications3a) In general, all reviewers appreciate that the paper was written for a general audience. However, please include all equations and note in the main text where they exist. For example, please include the equation for Communicability and expand the explanation – is it all walks, or all walks that are part of all shortest walks between all nodes? It is not clear in the Results or Materials and methods section.

We apologize for the lack of clarity. Communicability was calculated on all walks. This information (along with the equation defining this metric) has been added to the Materials and methods section.

The following text was added to the Materials and methods section:

“The communicability, C, between a pair of nodes i and j is defined as a weighted sum of the number of all walks connecting the pair of nodes (within weighted connectivity matrix, A) and has been shown to be equivalent to the matrix exponent of a binarized graph, e^A^ (Estrada and Hatano, 2008). For ease of interpretation, we calculated the log_10_-transformed mean of communicability for each graph across iterations and values of neural gain.”

Cij=∑k=0∞Akijk!=eA[8]

3b) Please unpack the rationale behind using the correspondence between structural and functional connectivity as criterion for delineating the "plausible" sub-space of parameter combinations. This is important because work into neuromodulation in small circuits shows that neuromodulation can reconfigure circuits, thus overriding the structural connectome (Marder, Neuron, 2012).

In light of the referenced paper, we have opted to remove the constraint that the structural connectome should be similar to the functional connectome. Importantly, this does not change the interpretation of our results.

3c) It seems that Vi is used for the computation of phase synchrony (i.e., not the neural activity passed through the hemodynamics). This should be made explicit in the Materials and methods. (Terms like "raw signal" or "neural data" are ambiguous.)

We have amended each instance of these terms in the manuscript to improve clarity.

3d) Figure 2 is described as inverted U relationship, a description that glosses over the left flank of this plot. The relationship clearly is monotonic, but not an inverted U.

We have added a statement to this effect in the manuscript.

3e) Figure 2—figure supplement 3 is missing axis labels and won't be understandable for a broad audience. Please explain what it shows and what that means.

The figure legends for Figure 2—figure supplement 3 have been amended in the updated version.